# Impact of different heat wave definitions on daily mortality in Bandafassi, Senegal

**Mbaye Faye** [1] *, **Abdoulaye Dème**[2], **Abdou Kâ Diongue**[1], **Ibrahima Diouf**[3]

1 LERSTAD—UFR Sciences Appliquées et de Technologies, Université Gaston Berger de Saint-Louis, Saint-Louis, Sénégal, 2 LSAO—UFR Sciences Appliquées et de Technologies, Université Gaston Berger de Saint-Louis, Saint-Louis, Sénégal, 3 NOAA Center for Weather and Climate Prediction Climate Prediction Center College Park, Maryland, United States of America

* faye.mbaye@ugb.edu.sn

## Abstract

### Objective

The aim of this study is to find the most suitable heat wave definition among 15 different ones and to evaluate its impact on total, age-, and gender-specific mortality for Bandafassi, Senegal.

### Methods

Daily weather station data were obtained from Kedougou situated at 17 km from Bandafassi from 1973 to 2012. Poisson generalized additive model (GAM) and distributed lag non-linear model (DLNM) are used to investigate the effect of heat wave on mortality and to evaluate the nonlinear association of heat wave definitions at different lag days, respectively.

### Results

Heat wave definitions, based on three or more consecutive days with both daily minimum and maximum temperatures greater than the 90[th] percentile, provided the best model fit. A statistically significant increase in the relative risk (RRs 1.4 (95% Confidence Interval (CI): 1.2–1.6), 1.7 (95% CI: 1.5–1.9), 1.21 (95% CI: 1.08–1.3), 1.2 (95% CI: 1.04–1.5), 1.5 (95% CI: 1.3–1.8), 1.4 (95% CI: 1.2–1.5), 1.5 (95% CI: 1.07–1.6), and 1.5 (95% CI: 1.3–1.8)) of total mortality was observed for eight definitions. By using the definition based on the 90[th] percentile of minimum and maximum temperature with a 3-day duration, we also found that females and people aged $\geq$ 55 years old were at higher risks than males and other different age groups to heat wave related mortality.

### Conclusion

The impact of heat waves was associated with total-, age-, gender-mortality. These results are expected to be useful for decision makers who conceive of public health policies in Senegal and elsewhere. Climate parameters, including temperatures and humidity, could be used to forecast heat wave risks as an early warning system in the area where we conduct

**Data Availability Statement:** The authors cannot provide the house location data and other identifying information, which would be against the ethical agreement with participants. However, for risk factor data analyses the data are fully available.

All additional data can be made available by contacting the authors Mbaye Faye (faye. mbaye@ugb.edu.sn) and/or Abdou Kâ Diongue (abdou.diongue@ugb.edu.sn).

**Funding:** This research was funded by ACASIS project (http://www.agence-nationale-recherche.fr/ Projet-ANR-13-SENV-0007). ACASIS project had access to the Bandafassi Health and Demographic Surveillance System (HDSS) and provided us daily mortality count data, but had no other role in the study design, data collection and analysis, decision to publish, or preparation of the manuscript.

**Competing interests:** The authors have declared that no competing interests exist.

this research. More broadly, our findings should be highly beneficial to climate services, researchers, clinicians, end-users and decision-makers.

## Introduction

Climate change poses major challenges to public health. A heat wave is an extreme weather phenomenon in meteorology that can directly or indirectly affect human health [1, 2]. The impact of heat waves on human health caused 70,000 excess deaths in Europe during the month of August 2003 [3, 4], 696 excess deaths in July 1995 in Chicago [5], 55,000 excess deaths in Russia in the summer of 2010 [6]. Three heatwaves occurred in Brisbane, Australia (January 2000; December 2001 and February 2004) and 51,233 deaths were recorded during the whole study period [7]. The Pakistan extreme heat wave of 2017 caused deaths of thousands of people [8]. A recent study in Istanbul, Turkey, suggests that the excess deaths, estimated at 419, were observed during the three heat wave episodes in 2015, 2016, and 2017 with increased risk of 11%, 6% and 21% respectively for each heatwave [9].

While heat waves that impact public health have been widely addressed in developed countries especially since the deadly heatwave that hit western Europe during the summer of 2003, no effort has been made to detect them and evaluate their impacts in developing countries, particularly in Africa where climate is warmer. However, the impact of climate change varies across many parts in Africa [10, 11], especially in Sub-Saharan Africa, whose adaptation capacities are low. Previous research has documented the impact of various definitions of heat wave on mortality in developed countries, unlike in developing countries where only a few studies have focused on single cities [12, 13]. Although heat wave definitions vary across world, as this definition depends on climatic zone, duration of heat wave, and metric of temperature and humidity and even winds in a lesser extent, some invariable metrics remain. To quantitatively reflect a heat wave event, the definition of heat wave should be complemented by the characterization of the following four metrics: (1) Magnitude: it should be computed based on an index or a set of indices of thermal condition(s) exceeding certain threshold(s), (2) Duration: which involves the computation of the persistence of a heat wave and should be based on recording the start time and the end time of the event, (3) Severity: it is a measurement method which integrates two aspects of the event, its magnitude and its persistence, and (4) Extent: it is computed to inform on the geographical area affected and the widespread the heat wave. Few researchers use hot days and nights [14, 15] as metrics of temperature, to define heat wave; they propose relative threshold or absolute threshold (fixed threshold) [16, 17]. Another study, conducted in West Africa, has used temperature $\geq 90^{th}$ percentile of both minimum and maximum temperature [18]. Some other studies have considered different types of definitions of heat wave and have examined the impact of heat waves on public health particularly and on mortality [19, 20]. A previous study used the Akaike Information Criterion (AIC) to determine the most appropriate definition of heat wave and identified the people who were sensitive to the first, second, and third heat wave [21]. A number of studies investigated cause-, age-, or gender-specific and heat-related mortality relationships, and found that the elderly subpopulation was more vulnerable and sensitivity by gender differed by region [22, 23]. Additionally, many previous studies showed that the relative risk mortality was higher for females [19, 20, 24, 25] and the identified the elderly as the most vulnerable group to heat wave [20, 26]. Many studies identified, various time lag effects [27, 28], and revealed that the relation between heat wave and mortality appeared immediately (lag0) [16] or was observed after some delay. The study periods used differs around the world and includes hot seasons, summer season, warm

season, or the whole year. In this study, we used the duration of heat waves and tested the percentile threshold approach. Different studies have applied various methods, including time-series analyses [29], both time-series analyses and meta-analysis [16], both systematic review and meta-analysis [30], case-crossover [31], Bayesian hierarchical models [19], and distributed lag non-linear model [32], quantile regression forests [28], and general circulation models [33]. A recent international scale study, using different heat wave definitions, found that people who live in moderate cold and moderate hot areas seem to be more susceptible to heat-related mortality than people who live in cold and hot areas [16]. However, a heat wave generally corresponds to a prolonged period of particularly high or extreme temperatures. Most studies use the daily maximum temperature [28, 33–37] or separately analyze daily minimum and maximum temperatures [38–42]. As highlighted by Perkins (2015) [43] these definitions have some common characteristics: temperature is always used in a raw or processed form, most often combined with a percentile-type threshold, and often based on a minimum duration of the heat wave.

In this present study, we aimed to use fifteen types of heat wave definitions (using both minimum and maximum temperature as an indicator), to identify an appropriate definition of heat wave, and to evaluate the impact of heat waves on mortality in Bandafassi, Senegal. Even these definitions are not specifically designed for health impacts, the detected heat waves are expected to be dangerous for public health. Currently, heat and-health warning systems do not exist at city level in Senegal. However, the results of this study are also aimed at developing the first heat warning system to reduce the heat related health exposure in Bandafassi.

The paper is structured as follows. Section 2 deals with the materials and methods. Section 3 delienates the results. Section 4 is devoted to the discussion of the findings, followed by a conclusion in section 5.

## Materials and methods

### Ethics statement

The full name of the institutional review board that reviewed our specific study for approval is the Bandafassi Health and Demographic Surveillance System (HDSS). We confirm that this study was approved.

### Study area

The Bandafassi area is located in Senegal, at 12.53˚ N, 12.32˚ W, with altitude ranging from 60 m to 426 m above the sea level mean. It is located in the region of Kedougou, in Eastern Senegal (Fig 1), near the border between Senegal, Mali and Guinea. The Bandafassi area is about 25 km long by 25 km large and with a total area of 600 km$^2$. It sits within the Sudanian savanna ecological zone. The climate is characterized by two seasons: a rainy season, from June to October, and a dry season, from November to May, with an average of rainfall of about 1,097 mm per year during the period 1984–1995. In Bandafassi, the highest minimum temperature and the maximum temperature generally are 36˚C and 49˚C respectively. The Bandafassi area is about 500 km distant from the capital Dakar. Kedougou is one of the warmest regions of the country during the dry season, with a bimodal annual cycle of temperature. During the rainy season, the temperature decreases significantly due to the landsurface cooling and the cloudy conditions associated with the large amounts of precipitation. Its climate ranges between Sudano-Sahelian and Sudano-Guinean. The peak rainfal is recorded in August; it can reach up to 200 mm during this month. The rainy season lasts 4 to 5 months, and it usually starts in May or June and ends in October [44]. The relative humidity is consistently high during the rainy season, peaking between August and October. However, the description is based on African meteorologists' practices which meets Köppens's classification [45].

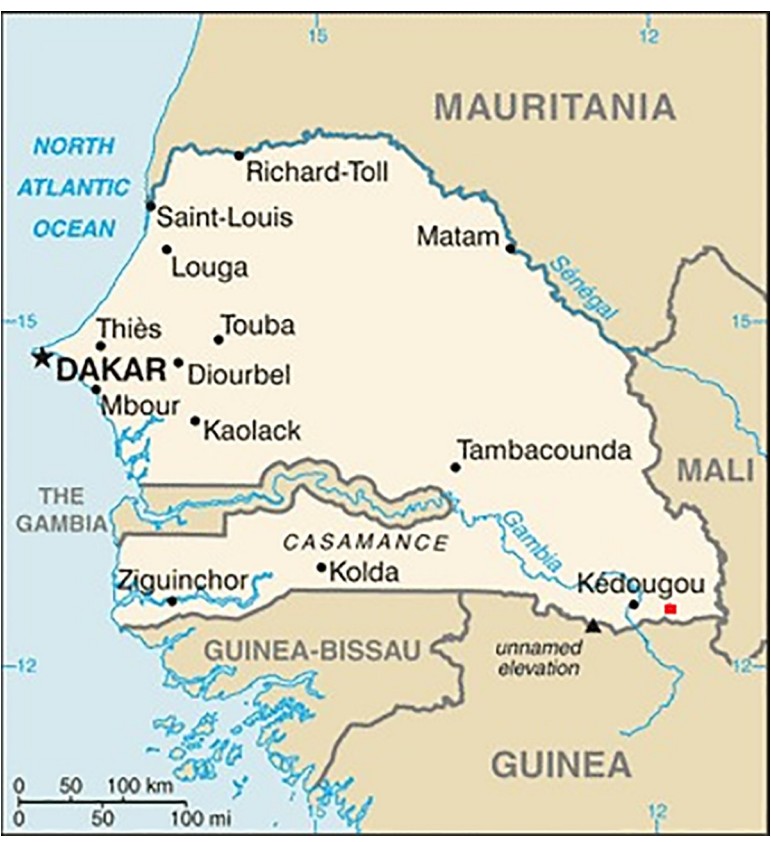

**Fig 1. Meteorological observation station.** The map shows the location of the station used in this study. We have marked the red square to highlight Bandafassi station. (https://www.cia.gov/the-world-factbook/).

## Data collection

We collected daily weather station data from Kedougou situated at 17 km from Bandafassi. The variables included daily minimum, maximum, and mean temperatures (°C), dew point temperature (°C), wind speed (m/s), and precipitation (mm) from 1973 to 2012 period.

We obtained daily mortality count data from the Bandafassi Health and Demographic Surveillance System (HDSS) for the same period, and included the date of death, age, gender and cause of death classified by the 10th Revision of the International Classification of Disease (ICD 10) code. We stratified mortality into two different groups based on gender (male and female) and age (0–5, 6–54, and ≥ 55 years old).

Our study has some limitations. Firstly, one rural area was considered, which suggests avoiding to generalizing the proposed heat wave definition in this study to other geographic areas. Secondly, we did not fully classify nor clarify the cause-specific deaths in our data. Air pollution is not available in our database that is why we did not take this variable into account through this study, and this is consistent with one previous study [46].

## Heat wave definition

There is no worldwide consensus on a heat wave definition, although several studies have proposed various definitions for heat waves based on metrics and threshold temperatures, and durations [13, 20]. Globally, heat wave definitions are based on the temperature metrics, absolute or relative temperature threshold within consecutive days.

We combined both daily minimum and maximum temperature [8, 47–51] as an indicator to define a heat wave and to assess the impact on mortality. Heat wave definition is characterized by intensity, frequency, duration, and timing in the season [2, 15, 21, 51, 52]. In the scientific literature, previous studies have used fifteen types of definitions of heat waves depending on diverse temperature metrics (e.g., minimum, maximum or mean temperature, or apparent temperature), duration ($\geq 2$, $\geq 3$ and $\geq 4$), and absolute or relative temperature threshold (87th, 90th, 92th, 95th, 97th, 98th, 99th percentile) [21, 53].

In this study, we used the fifteen types of definitions of heat waves by combining relative thresholds (87th, 90th, 92th, 95th, 97th percentile of both daily minimum and maximum temperature) and duration ($\geq 3$, $\geq 4$ and $\geq 5$) (Table 1).

Table 1 provides the different duration and intensity. The heat wave day of the 87th temperature percentile with duration $\geq 3$ days in Bandafassi is higher than the combination of the other durations and intensities. We observed that, if the intensity is high, the duration increases, the number of heat wave days decreases as well.

## Statistical analysis

We also used the Poisson generalized additive model (GAM) [54] to assess the effect of heat wave on mortality. The daily mortality count follows the Poisson distribution model. Relative risks (RRs) and 95% confidence intervals (CIs) were calculated using GAMs. The GAM model was given as follows:

$$\ln E(Y_t) = \beta + S(DOY_t) + DOW_t + Year_t + S(T_t, df) + HW_t + \varepsilon_t \qquad (1)$$

$E(Y_t)$ is the expected daily mortality counts on day t, $\beta$ is the intercept, $DOY_t$ represents day of year, $DOW_t$ is the categorical variable for day of the week, $Year_t$ represents a long-term trend, $T_t$ is a temperature metric for a specific lag from a lag day t, the degree of freedom (df) in the spline smoothing function of temperature was 5 according to Akaike Information Criterion (AIC) [55], $HW_t$ is a binary variable, which equals to 0 for non-heat wave days and 1 for heat wave days, for under different heat wave definition, $\varepsilon_t$ is the statistical error, t is the day of observation, and S() denotes the cubic smoothing spline.

**Table 1. Summary statistics of heat wave days based on different duration ($\geq 3$, $\geq 4$, $\geq 5$ days) and intensity (87th, 90th, 92th, 95th, 97th), and the total number of days from 1973 to 2012.**

| Heat wave name | Heat wave definition | Heat wave days |
| --- | --- | --- |
| HWD_87P_3day | 87th percentile with 3 days duration | 158 |
| HWD_87P_4day | 87th percentile with 4 days duration | 105 |
| HWD_87P_5day | 87th percentile with 5 days duration | 70 |
| HWD_90P_3day | 90th percentile with 3 days duration | 95 |
| HWD_90P_4day | 90th percentile with 4 days duration | 55 |
| HWD_90P_5day | 90th percentile with 5 days duration | 32 |
| HWD_92P_3day | 92th percentile with 3 days duration | 75 |
| HWD_92P_4day | 92th percentile with 4 days duration | 41 |
| HWD_92P_5day | 92th percentile with 5 days duration | 22 |
| HWD_95P_3day | 95th percentile with 3 days duration | 16 |
| HWD_95P_4day | 95th percentile with 4 days duration | 5 |
| HWD_95P_5day | 95th percentile with 5 days duration | 2 |
| HWD_97P_3day | 97th percentile with 3 days duration | 4 |
| HWD_97P_4day | 97th percentile with 4 days duration | 0 |
| HWD_97P_5day | 97th percentile with 5 days duration | 0 |

The Distributed Lag Non-linear Model (DLNM) was used to evaluate the nonlinear association of heat wave definition at different lag days [56, 57]. A maximum lag of 25 days was utilized as sufficient length of time to simultaneously estimate of the non-linear and delayed effects of heat wave on mortality. The DLNM model used a "cross-basis" function, which allows simultaneous estimation of the non-linear effects across lag [58]. A natural spline cubic DLNM was adopted to capture the non-linear relationship between the covariate and dependent variable. The model structure is as follows:

$$Y_t \sim \text{Poisson}\ (\mu_t)$$

$$\ln(\mu_t) = \alpha + \text{ns}(\text{DOY}_t) + \text{DOW}_t + \text{ns}(\text{Year}_t) + \text{ns}(T_t, df = 5) + \text{cb}(\text{HW}_t, \text{lag}) + \varepsilon_t \quad (2)$$

where t denotes the day of the observation, $Y_t$ is the number of deaths on day t, $\mu_t$ is the mean mortality count for day t, $\alpha$ denotes intercept term, $\text{DOY}_t$ means the day of year, $\text{DOW}_t$ is the categorical variable for day of the week, $\text{Year}_t$ represents a long-term trend, $T_t$ is a temperature metric for a specific lag from a lag day t, the degree of freedom (df) is chosen by the Akaike Information Criterion (AIC), $\text{HW}_t$ represents the heat wave on day t, $\varepsilon_t$ is the statistical error, ns() denotes the natural cubic spline, and cb() means cross-basis function.

R software version 3.2.2, with the mgcv and the "dlnm" package [59], was used for all statistical analysis and figures.

## Results

Table 2 provides the summary of the statistic of the demographic characteristic of daily mortality and meteorological factors in Bandafassi. During the study period 1973–2012, the total number was 6,684 and the average daily all, male, female, 0–5 years, 6–54 years, 55 and more years mortality count were 0.27, 0.14, 0.12, 0.078, 0.046, 0.15, respectively. In the same period, the average daily minimum, mean, maximum temperature were 22.7°C (range from 10°C to 36°C), 29.5°C (range from 16.1°C to 39.8°C), 35.4°C (range from 17.5°C to 49°C), respectively.

From Fig 2, we observed the daily number of deaths and the daily minimum and maximum temperatures during the four-decade period (1973–2012) in Bandafassi. The mortality data depicted five major peaks, the first one appeared on the 27th April 1980, the second is observed on the 4th May 2005, the third is obtained on the 10th May 2006, the fourth occurred on the 6th March 2010 and the last one became visible on the 17th April 2010.

**Table 2. Summary demographic characteristic daily mortality and meteorological factors in Bandafassi, Senegal (1973–2012).**

| Variables | Mean (SD) | Min | Max |
|---|---|---|---|
| **Demographic characteristic** | | | |
| All | 0.27 (1.79) | 0 | 21 |
| Male | 0.14 (0.96) | 0 | 12 |
| Female | 0.12 (0.85) | 0 | 11 |
| 0–5 years | 0.078 (0.54) | 0 | 7 |
| 6–54 years | 0.046 (0.30) | 0 | 4 |
| 55+ years | 0.15 (1.06) | 0 | 13 |
| **Meteorological factor** | | | |
| Minimum temperature (°C) | 22.7 (3.1) | 10 | 36 |
| Mean temperature (°C) | 29.5 (2.7) | 16.1 | 39.8 |
| Maximum temperature (°C) | 35.4 (3.1) | 17.5 | 49 |

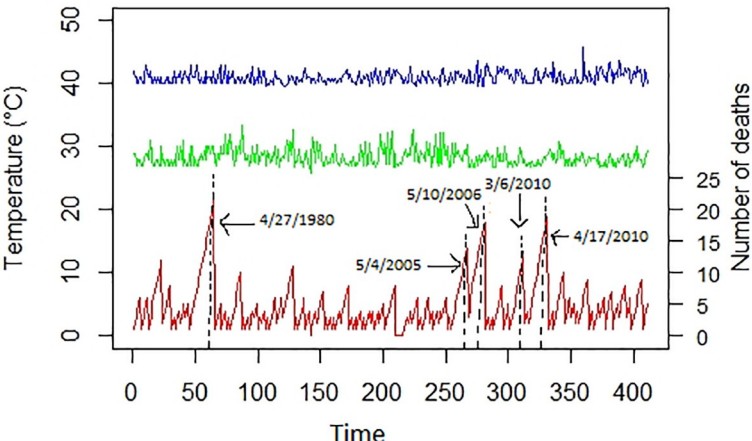

**Fig 2.** The daily number of deaths (red line), the daily minimum temperature (green line), and the daily maximum temperature (blue line) during the period 1973–2012 in Bandafassi.

Table 3 displays the sum of AIC values of all gender- and age-specific mortality for different heat wave definitions. In Bandafassi, the heat wave defined by threshold of 90th percentile of temperature with duration ≥3 days produced the lowest AIC value 44,660.4.

Table 4 gives the relative risk of heat wave days based on 15 different heat wave definitions in Bandafassi. Eight heat wave definitions (90th, 92th percentile with duration ≥ 3 days, 87th, 90th, 92th percentile with duration ≥ 4 days, and 87th, 90th, 92th percentile with duration ≥ 5 days) were statistically significant in relating total mortality to the relative risk of 1.4 (95% CI: 1.2–1.6), 1.7 (95% CI: 1.5–1.9), 1.21 (95% CI: 1.08–1.3), 1.2 (95% CI: 1.04–1.5), 1.5 (95% CI: 1.3–1.8), 1.4 (95% CI: 1.2–1.5), 1.5 (95% CI: 1.07–1.6), and 1.5 (95% CI: 1.3–1.8) for total mortality, respectively. Seven heat wave definitions (90th, 92th, 95th percentile with duration ≥ 3 days, 87th, 92th percentile with duration ≥ 4 days, and 87th, 90th percentile with duration ≥ 5 days) statistically increased the risk of female mortality given by 1.609 (95% CI: 1.33–1.88), 1.72 (95% CI: 1.45–2.005), 2.87 (95% CI: 1.9–3.8), 1.21 (95% CI: 1.039–1.39), 1.49 (95% CI: 1.14–1.83), 1.37 (95% CI: 1.16–1.59), and 1.54 (95% CI: 1.15–1.92), respectively. Seven heat wave definitions (90th, 92th, 95th percentile with duration ≥ 3 days, 87th, 92th percentile with duration ≥ 4 days, and 87th, 90th percentile with duration ≥ 5 days) significantly associated male mortality with the relative risk of 1.39 (95% CI: 1.11–1.67), 1.87 (95% CI: 1.58–2.16), 3.8 (95% CI: 2.8–4.8), 1.21 (95% CI: 1.037–1.39), 1.74 (95% CI: 1.37–2.105), 1.47 (95% CI: 1.26–1.69), and 1.709 (95% CI: 1.28–2.13), respectively. For age category-specific (0–5 years), six heat wave definitions (90th, 92th, 95th percentile with duration ≥ 3 days, 87th, 92th percentile with duration ≥ 4 days, and 87th percentile with duration ≥ 5 days) were statistically

**Table 3. Value of the sum of the Akaike Information Criterion for Poisson (AIC) of heat wave days based on different duration (≥3, ≥4, ≥5 days) and intensity (87th, 90th, 92th, 95th, 97th) in Bandafassi for years 1973–2012.**

| Heat wave threshold (percentile of temperature) | Value of AIC | | |
|---|---|---|---|
| | | 3 days | 4 days | 5 days |
| 87th | | 44,676.5 | 44,662.3 | 44,664.3 |
| 90th | | **44,660.4** | 44,670.2 | 44,677.2 |
| 92th | | 44,670.9 | 44,676.4 | 44,676.0 |
| 95th | | 44,675.5 | 44,678 | 44,677.1 |
| 97th | | 44,675.8 | 44,676.0 | 44,676.0 |

**Table 4. Relative Risk (RR) of daily mortality during heat wave based in different duration (≥3, ≥4, ≥5 days) and intensities (87th, 90th, 92th, 95th, 97th) during the period 1973–2012.**

| Mortality | 3 days | 4 days | 5 days |
|---|---|---|---|
| | RRs 95%CI | RRs 95%CI | RRs 95%CI |
| **Total** | | | |
| 87th | 1.005 (0.9–1.1) | 1.21 (1.08–1.3)** | 1.4 (1.2–1.5)*** |
| 90th | 1.4 (1.2–1.6)*** | 1.2 (1.04–1.5)* | 1.5 (1.07–1.6)** |
| 92th | 1.7 (1.5–1.9)*** | 1.5 (1.3–1.8)*** | 1.5 (1.3–1.8)*** |
| 95th | 2.9 (2.2–3.5) | 1.2 (-0.27–2.8) | 0.1 (-2.8–3.1) |
| 97th | --- --- | --- --- | --- --- |
| **Female** | | | |
| 87th | 1.0013 (0.86–1.14) | 1.21 (1.039–1.39)* | 1.37 (1.16–1.59)** |
| 90th | 1.609 (1.33–1.88)*** | 1.38 (1.051–1.71) | 1.54 (1.15–1.92)* |
| 92th | 1.72 (1.45–2.005)*** | 1.49 (1.14–1.83)* | 1.34 (0.9–1.78) |
| 95th | 2.87 (1.9–3.8)* | 0.68 (-3.23–4.61) | 0.17 (-4.7–5.1) |
| 97th | --- --- | --- --- | --- --- |
| **Male** | | | |
| 87th | 1.011 (0.87–1.14) | 1.21 (1.037–1.39)* | 1.47 (1.26–1.69)*** |
| 90th | 1.39 (1.11–1.67)* | 1.22 (0.87–1.57) | 1.709 (1.28–2.13)* |
| 92th | 1.87 (1.58–2.16)*** | 1.74 (1.37–2.105)** | 1.52 (1.08–1.97) |
| 95th | 3.8 (2.8–4.8)** | 4.056 (1.7–6.41) | 0.17 (-4.7–5.1) |
| 97th | 1.46 (0.05–2.88) | --- --- | --- --- |
| **0–5 years** | | | |
| 87th | 1.029 (0.9–1.15) | 1.21 (1.039–1.38)* | 1.45 (1.24–1.66)*** |
| 90th | 1.44 (1.17–1.7)** | 1.2 (0.87–1.53) | 1.4 (1.001–1.85) |
| 92th | 1.86 (1.58–2.13)*** | 1.56 (1.21–1.91)* | 1.29 (0.8–1.74) |
| 95th | 5.1 (4.1–6.2)** | 0.56 (-6.5–7.6) | --- --- |
| 97th | --- --- | --- --- | --- --- |
| **6–54 years** | | | |
| 87th | 0.85 (0.65–1.064) | 1.051 (0.78–1.31) | 1.206 (0.902–1.51) |
| 90th | 1.405 (0.904–1.907) | 1.47 (0.89–2.057) | 1.74 (1.107–2.38) |
| 92th | 1.52 (1.037–2.018) | 1.68 (1.107–2.27) | 1.64 (0.98–2.29) |
| 95th | 1.31 (-0.69–3.31) | 2.61 (-0.34–5.57) | 0.62 (-3.44–4.69) |
| 97th | --- --- | --- --- | --- --- |
| **≥55 years** | | | |
| 87th | 1.12 (0.92–1.33) | 1.41 (1.15–1.66)** | 1.63 (1.32–1.94)** |
| 90th | 1.67 (1.31–2.031)** | 1.39 (0.96–1.82) | 1.86 (1.37–2.35)* |
| 92th | 1.87 (1.5–2.23)*** | 1.64 (1.19–2.097)* | 1.54 (0.95–2.35) |
| 95th | 2.37 (1.29–3.45) | 2.83 (0.13–5.54) | --- --- |
| 97th | --- --- | --- --- | --- --- |

--- = not enough data to generate a reliable estimate

***p-value < 0.001

**p-value < 0.01

*p-value < 0.05.

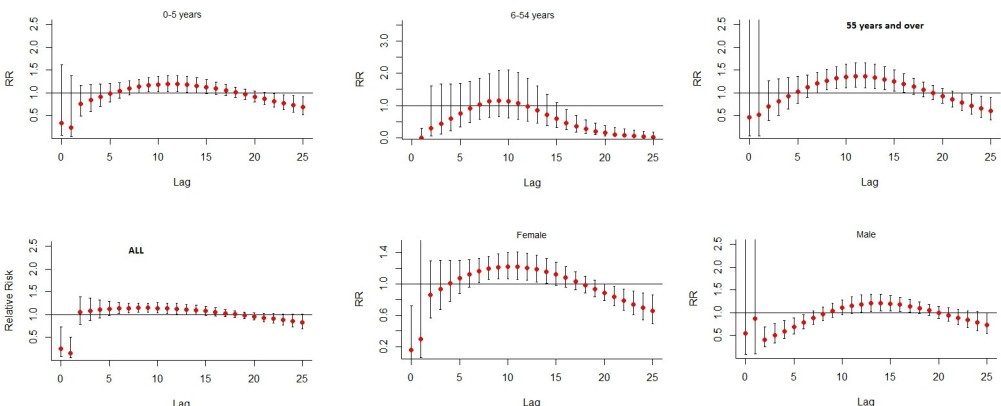

**Fig 3. Relative risk of mortality on the lag distribution of heat wave stratified by gender, and age based on the definition ≥90th percentile with duration ≥3 consecutive days as heat wave.**

significantly associated with the risk mortality of 1.44 (95% CI: 1.17–1.7), 1.86 (95% CI: 1.58–2.13), 5.1 (95% CI: 4.1–6.2), 1.21 (95% CI: 1.039–1.38), 1.56 (95% CI: 1.21–1.91), and 1.45 (95% CI: 1.24–1.66), respectively. For people over age of 55, we found that six heat wave definitions (90th, 92th, percentile with duration ≥ 3 days, 87th, 92th percentile with duration ≥ 4 days, 87th, and 90th percentile with duration ≥ 5 days) had significant risk of mortality as follows 1.67 (95% CI: 1.31–2.031), 1.87 (95% CI: 1.5–2.23), 1.41 (95% CI: 1.15–1.66), 1.64 (95% CI: 1.19–2.097), 1.63 (95% CI: 1.32–1.94), and 1.86 (95% CI: 1.37–2.35), respectively. The relative risk among females was higher than that found for males to heat wave related mortality for the definition based on 90th temperature percentile with 3-day (produced the best model fit as judged by AIC). No significant relative risk was observed among people aged 6–54 years for all heat wave definitions. For most of the cases, the highest mortality risk was observed for 95th temperature percentile with 3-day, and 4-day. For age-specific categories, the relative risk in the population for people over the age of 55 years was larger than that of the other group. The children people aged 0–5 years are at high risk of total mortality and gender-, age-specific mortality for the heat wave definition (95th temperature percentile with duration ≥ 3 days). The results reveal that the mortality risk associated with heat wave depends on both temperature threshold and duration.

Fig 3 reveals the mortality risk of heat wave definition using the 90th percentile of temperature with duration ≥ 3 days for various lagged days. We consider the lag distribution from lag 0 days to lag 25 days. The graph shows that the relative risk of all mortality, gender-, and age-specific was below 1.0 in lag 0 days. An increase of the relative risk was first observed, and it was followed by a decline as seen in Fig 3. We observed significant association for all mortality at lags 6–12 days, although the highest relative risk appeared at lag 8 days and lag 9 days respectively (RR = 1.11; 95% CI: 1.23–1.35 and RR = 1.13; 95% CI: 1.25–1.36). We found significant associations among male mortality at lags 11–18 days, and no effect thereafter. For female mortality, the associations significant occurred at lags 7–14 days. Among the children aged 0–5 years, there was statistically significant at lags 8–14 days. The people aged 55 years old or above were significantly higher risk of heat wave associated mortality at lags 7–16 days. We observed an insignificant associated for the people aged 6–54 years at different lags.

## Discussion

This is the first study which examines the impact of different heat wave definitions on mortality in Senegal as far as we know. Although, different definitions for heat wave have been

documented around the world, for example in America, Europe and Asia (especially in China). Zhang et al. (2017) [20] used 45 definitions of heat wave by combining five temperature thresholds and three temperature indicators (90th, 92.5th, 95th, 97.5th and 99th percentile of daily mean temperature, minimum temperature and maximum temperature) with duration ($\geq 2, \geq 3$ and $\geq 4$ days) to assess the impact of heat wave under different definitions on non-accidental mortality, and found that heat wave defined by daily mean temperature $\geq 99$th percentile and duration $\geq 3$ days showed the best model fit among the 46 heat wave definitions. Seposo et al. (2017) [32] developed 15 heat wave definitions combining different intensities (90th, 95th, 97th, 98th and 99th temperature percentile) with heat wave duration ($> 2, > 4$, and $> 7$ consecutive days) as heat wave candidates. In this study, we used 15 definitions of heat wave by combining temperature metrics (both temperature maximum and minimum), temperature threshold (87th, 90th, 92th, 95th, 97th percentile) and duration ($\geq 3, \geq 4$ and $\geq 5$ days) in Bandafassi during the period of 1973–2012. Based on these different heat wave definitions, the best model fit was produced by heat wave definition using both minimum and maximum temperature $\geq 90$th percentile with duration $\geq 3$ consecutive days (judged by Akaike Information Criterion (AIC)). This result is consistent with numerous previous studies such as [48].

To define a heat wave, many studies used relative threshold [16, 21] while others used only absolute threshold [12]. Similar methods were observed in this study. In the literature, we found numerous studies using both relative and absolute threshold. For example, Tong et al. (2010) [7] used 10 heat wave definitions, including both absolute and relatives ones [60], in contrast Chen et al. (2015) [53] identified 15 different heat wave definitions, including both relative and absolute threshold [53]. Our study used only relative threshold. The age at death was stratified into three groups which considered young children (0–5 years old), people aged from 6 to 54 years and elderly ($\geq 55$ years old).The age ranges are justified by Table 2 in the work of Pison and Langaney (1984) [61], they found that the probability of death decreases with age up to 5 years old then increases until around 54 years old and then increases again. In the literature, the age group differs, each study gives its own subgroup. Mortality risk varied significantly by age and gender between different heat wave definitions. We observed a small age-specific difference, the elderly ($\geq 55$ years old) are more fragile population than young children (0–5 years old). Even young children are potentially at great risk for heat wave as evidenced in a previous review [62]. We also found that the elderly ($\geq 55$ years old) were more susceptible to heat wave effects, which is consistent with earlier work [53]. Our findings show that the relative risk among 6–54 year age groups was statistically insignificant for the different heat wave definitions considered as shown in Table 4. For age-specific, the highest mortality risk was observed among elderly people at the heat wave definition using daily minimum temperature of 28°C ($\geq 95$th percentile) and daily maximum temperature of 40.72°C ($\geq 95$th percentile) with a duration $\geq 3$ days.

It was also found that the relative risk mortality among females were higher than for males in the heat wave definition based on 90th temperature percentile with 3-day and 4-day duration, and this is in line with a number of previous studies [22, 24]. In contrast, the relative risk mortality among males was higher than females by the definition combining threshold (87th, 92th, 95th, 97th percentile) with duration ($\geq 3, \geq 4$ and $\geq 5$ days) and 90th temperature percentile with 5-day duration. These results are consistent with some previous studies [32]. The effect of the heat on people living in Bandafassi is not immediate as evidenced in most countries worldwide but delayed after a few days. These results are in agreement with numerous previous studies [7, 16, 63, 64]. The reason is that, possibly due to their greater physiological adaptation to high temperatures. For gender-specific, the highest mortality risk appeared among males at the heat wave definitions using a daily minimum temperature of 28°C ($\geq 95$th

percentile) and daily maximum temperature of 40.72˚C ($\geq$ 95th percentile) with the duration $\geq$ 4 days. Under different heat wave definitions, certain extant studies used two or four days as the duration without proof of their choice. In our work, we chose 3-, 4-, or 5-days as the duration to define heat wave, because of the significance of the heat wave duration. As it is shown in Table 4, the 3-, 4-, and 5-days duration and 97th temperature percentile definition don't produce a reliable estimate because there is insufficient data except for the 3-day duration and 97th temperature percentile among male risk mortality.

In Fig 3, we observed the short-term mortality displacement also known as harvesting [63, 64]. The harvesting occurs when a positive association at short lags is followed by a negative association at longer lags which should suggest a 'deficit' of mortality [65]. In the 6 to 54-year age group, we observed negative mortality at lags 0–6 days, a positive association from lags 7–11 days followed by a negative association at lags 12–25 days. Among those 0–5 years of age, negative mortality occurred across lags of 0–5 days, we observed a positive association from lags 6–18 days followed by a negative association at lags 19–25 days. Among the people aged 55 years and over, a presence of negative mortality appeared from lags 0–4 days, a positive association from lags 5–18 days followed by a negative association at lags 19–25 days are shown. For males, we observed a clear decline in mortality around lags of 0–4 days, an increase in mortality across lags of 5–16 days followed by a decrease in mortality across lag of 17–25 days. For females, we observed negative mortality around lag of 0–8 days, a positive association around lags of 9–19 days followed by a negative association at lags 20–25 days. For all deaths, a decline in mortality occurred across a lag of 0–1 days, we found a positive association from lags 2–18 days followed by a negative association at lags 19–25 days. These results are displayed in Fig 3, which shows similar patterns to the results obtained by a previous study [66]. For heat wave definition, a number of previous studies applied a combination of intensity and duration [67, 68]. In this paper, we did not evaluate the cause-specific of deaths which is not compatible with other numerous previous studies [24, 69].

## Conclusions

Considering 15 definitions of heat wave, we found that using both the minimum and maximum temperatures threshold of $\geq$ 90th percentile with duration $\geq$ 3 days was the most suitable definition to capture the impact of heat wave on mortality in Bandafassi during the period of 1973–2012. The findings of our work show also that elderly people ($\geq$ 55 years old) are frail to heat-related mortality and females seem to be at higher risk than males to the mortality impact of heat wave definition based on 90th temperature percentile with 3-day in Bandafassi. For the sub-population between 6 and 54 years old, the relative risk values can be high, but they are not statistically significant, regardless of the analyzed percentile and heat wave duration. These results are expected to be useful for decision makers who plan public health measures in Senegal and elsewhere. Climate parameters including temperatures and humidity could be used to forecast heat wave risks over our area study for an early warning system. More broadly, our findings should be crucial for climate services, researchers, clinicians, end-users and decision-makers.

## Supporting information

**S1 Dataset.**
(CSV)

**S1 Fig. Relative Risk (RR) of mortality on the lag distribution of heat wave stratified by gender, and age based on the definition $\geq$ 90th percentile of apparent temperature with**

duration $\geq$ 3 consecutive days as heat wave.
(DOCX)

**S2 Fig. Annual cycle of Tmax (maximum temperature) and Tappmax (maximum apparent temperature) in Kedougou (1973–2012).**
(DOCX)

**S3 Fig. Annual cycle of Tmean (mean temperature) and Tapp (mean apparent temperature) in Kedougou (1973–2012).**
(DOCX)

**S4 Fig. Annual cycle of Tmin (minimum temperature) and Tappmin (minimum apparent temperature) in Kedougou (1973–2012).**
(DOCX)

**S1 Table. Displays the comparison (between apparent temperature and ambient temperature) of the sum of AIC values of all gender- and age- specific mortality for different heat wave definitions.** Ambient temperature is the best predicteur of mortality in our study in term of AIC because the results with ambient temperature produced the lowest AIC value.
(DOCX)

**S2 Table. Relative Risk (RR) of daily mortality during heat wave based in different duration ($\geq$3, $\geq$4, $\geq$5 days) and intensities (87th, 90th, 92th, 95th, 97th percentile of apparent temperature) during the period 1973–2012.**
(DOCX)

## Author Contributions

**Conceptualization:** Mbaye Faye.

**Software:** Mbaye Faye, Ibrahima Diouf.

**Supervision:** Abdou Kâ Diongue.

**Validation:** Abdou Kâ Diongue.

**Writing – original draft:** Mbaye Faye.

**Writing – review & editing:** Abdoulaye Dème, Abdou Kâ Diongue, Ibrahima Diouf.

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
