## [Decision Letter · Decision Letter 0]

8 Oct 2020

PONE-D-20-27586

Impact of Different Heat Wave Definition on Daily Mortality in Bandafassi, Senegal

PLOS ONE

Dear Dr. Mbaye,

Thank you for submitting your manuscript to PLOS ONE. After careful consideration, we feel that it has merit but does not fully meet PLOS ONE’s publication criteria as it currently stands. Therefore, we invite you to submit a revised version of the manuscript that addresses the points raised during the review process.

Both the reviewers are in favour to publish the article. However, both of them asked for few revisions. Though both of them asked for minor revision, reviewer 2 seems critical on some issues. I also agree with reviewer 2 that the literature review is not complete. Besides, author should try to highlight the novelty of the study.

We look forward to receiving your revised manuscript.

Kind regards,

Shamsuddin Shahid

Academic Editor

PLOS ONE

Additional Editor Comments:

Both the reviewers are in favour to publish the article. However, both of them asked for few revisions. Though both of them asked for minor revision, reviewer 2 seems critical on some issues. I also agree with reviewer 2 that the literature review is not complete. Besides, author should try to highlight the novelty of the study.

Journal Requirements:

"This research was funded by ACASIS project (http://www.agence-nationalerecherche.fr/Projet-ANR-13-SENV-0007). Authors also thank Serge Janicot and Richard Lalou

for useful discussions about climate health impacts and their support through ACASIS

project."

3. Please ensure that you refer to Figure 1 in your text as, if accepted, production will need this reference to link the reader to the figure.

4. We note that Figure 1 in your submission contain map images which may be copyrighted. All PLOS content is published under the Creative Commons Attribution License (CC BY 4.0), which means that the manuscript, images, and Supporting Information files will be freely available online, and any third party is permitted to access, download, copy, distribute, and use these materials in any way, even commercially, with proper attribution. For these reasons, we cannot publish previously copyrighted maps or satellite images created using proprietary data, such as Google software (Google Maps, Street View, and Earth). For more information, see our copyright guidelines: http://journals.plos.org/plosone/s/licenses-and-copyright.

4.1.    You may seek permission from the original copyright holder of Figure 1 to publish the content specifically under the CC BY 4.0 license. 

4.2.    If you are unable to obtain permission from the original copyright holder to publish these figures under the CC BY 4.0 license or if the copyright holder’s requirements are incompatible with the CC BY 4.0 license, please either i) remove the figure or ii) supply a replacement figure that complies with the CC BY 4.0 license. Please check copyright information on all replacement figures and update the figure caption with source information. If applicable, please specify in the figure caption text when a figure is similar but not identical to the original image and is therefore for illustrative purposes only.

Reviewers' comments:

Reviewer's Responses to Questions

**Comments to the Author**

1. Is the manuscript technically sound, and do the data support the conclusions?

Reviewer #1: Yes

Reviewer #2: Yes

2. Has the statistical analysis been performed appropriately and rigorously? 

Reviewer #1: Yes

Reviewer #2: I Don't Know

3. Have the authors made all data underlying the findings in their manuscript fully available?

Reviewer #1: No

Reviewer #2: No

4. Is the manuscript presented in an intelligible fashion and written in standard English?

Reviewer #1: Yes

Reviewer #2: Yes

5. Review Comments to the Author

Reviewer #1: Review Report 1 for manuscript “PONE-D-20-27586”

I have read this manuscript with great interest. The study tries to select the most suitable heat wave definition for Bandafassi, Senegal. I think this paper will please the readers of PLOS one. It used very simple methodologies. However, the discussion should be improved. It is more like a literature review. My recommendation is minor revision. Below are my comments.

L17: please, replace “the best” by “the most suitable” and add “for Bandafassi” at the end of the objective.

L26: What is CI? The reader don’t know yet its meaning yet.

L31: What is “with 3-day”? Do you mean with 3-day duration?

L37: remove “current”

L41:43: please rewrite starting from “three heat waves …. Study periods”. The sentence is misleading. Also in line 43 and 44. I think it should be “A recent study…. summaries that the excess ….. during the three heat …… 2017) showed a 11% ….. increase in risk.”

L60: start time and end time.

L119: What do you refer to in this sentence?

L135:136: move to statistical analysis

L202: coma is missing “from figure 2, we “

Figure 2: It is better to re arrange the x-axis to incremental values and add these specific days as vertical lines (using abline in R).

L215: Bold is not needed.

Page 12: Please, divide into several paragraphs.

L323: Is it the only reason? Is it or due to heat built-up in body?

L344: correct to “3 days is the most suitable (OR most representative) definition”

Good Luck

Reviewer #2: The heat wave have highest morality rate in cities while this paper only explores the rural heat wave morality. The reason is that heat wave is amplified a lot when considering the cities. I suggest the results from the cities should also be considered.

What was the base year the heat wave was defined?

The heat wave is indeed defined by a temperature, but it does not mean that the threshold temperature should be percentage based. The question is why the percentage-based threshold was used, why not a specific temperature?

Why only morality data was used why also use the morbidity data also?

What is the correlation coefficient in figure 2? I think it will give some idea between the temperature and death perhaps?

What is the novelty of the study?

The literature is not complete. Please include the papers on heat wave definition, indices and what heat waves will be in the coming decades:

-Trends in heat wave related indices in Pakistan

-Prediction of heat waves in Pakistan using quantile regression forests

-Spatial distribution of unidirectional trends in temperature and temperature extremes in Pakistan

-Selection of GCMs for the projection of spatial distribution of heat waves in Pakistan

6. PLOS authors have the option to publish the peer review history of their article (what does this mean?). If published, this will include your full peer review and any attached files.

Reviewer #1: **Yes: **Mohamed Salem Nashwan

Reviewer #2: No

---

## [Author Response · Author response to Decision Letter 0]

4 Jan 2021

November, 2020

Re: Resubmission of “ Impact of Different Heat Wave Definition on Daily Mortality in Bandafassi, Senegal”, manuscript id: PONE-D-20-27586 

Dear Editor :

Thank you for the opportunity to revise our manuscript titled “ Impact of Different Heat Wave Definition on Daily Mortality in Bandafassi, Senegal”. We appreciate the careful reviews and constructive suggestions by the reviewers. The manuscript has substantially improved after making the suggested amendments.

In the following section, find a detailed point-by-point response in red to the reviewers and the editor’s concerns. Changes made in the manuscript are marked using track changes. The revision has been developed in consultation with all co-authors, and each author has given approval to the final draft.

Sincerely, 

Mbaye Faye

Saint-Louis, Senegal

Reponses to Editor’s comments

Comment 1: Please ensure that your manuscript meets PLOS ONE's style requirements, including those for file naming. The PLOS ONE style templates can be found at

Answer 1: We have updated and re-structured the text as required. The manuscript has been edited according to the above style guidelines to fit PLOS ONE's style requirements.

Comment 2: Thank you for stating the following in the Acknowledgments Section of your manuscript:

"This research was funded by ACASIS project (http://www.agence-nationalerecherche.fr/Projet-ANR-13-SENV-0007). Authors also thank Serge Janicot and Richard Lalou for useful discussions about climate health impacts and their support through ACASIS project."

Answer 2: We have removed “This research was funded by ACASIS project (http://www.agence-nationalerecherche.fr/Projet-ANR-13-SENV-0007). Authors also thank Serge Janicot and Richard Lalou for useful discussions about climate health impacts and their support through ACASIS project” as suggested.

Comment 3: Please ensure that you refer to Figure 1 in your text as, if accepted, production will need this reference to link the reader to the figure.

Answer 3: Thank you for noting this. This is done in the study area subsection (L115).

Comment 4: We note that Figure 1 in your submission contain map images which may be copyrighted. All PLOS content is published under the Creative Commons Attribution License (CC BY 4.0), which means that the manuscript, images, and Supporting Information files will be freely available online, and any third party is permitted to access, download, copy, distribute, and use these materials in any way, even commercially, with proper attribution. For these reasons, we cannot publish previously copyrighted maps or satellite images created using proprietary data, such as Google software (Google Maps, Street View, and Earth). For more information, see our copyright guidelines: http://journals.plos.org/plosone/s/licenses-and-copyright.

4.1. You may seek permission from the original copyright holder of Figure 1 to publish the content specifically under the CC BY 4.0 license. 

4.2. If you are unable to obtain permission from the original copyright holder to publish these figures under the CC BY 4.0 license or if the copyright holder’s requirements are incompatible with the CC BY 4.0 license, please either i) remove the figure or ii) supply a replacement figure that complies with the CC BY 4.0 license. Please check copyright information on all replacement figures and update the figure caption with source information. If applicable, please specify in the figure caption text when a figure is similar but not identical to the original image and is therefore for illustrative purposes only.

Answer 4: Thank you for your suggestion, we replace the previous figure by one that complies with the CC BY 4.0 license.

Reponses to Reviewer #1 comments

I have read this manuscript with great interest. The study tries to select the most suitable heat wave definition for Bandafassi, Senegal. I think this paper will please the readers of PLOS one. It used very simple methodologies. However, the discussion should be improved. It is more like a literature review. My recommendation is minor revision. Below are my comments.

We appreciate the reviewer’s comments. As the reviewer suggested, we have revised the discussion section in the update manuscript. We will improve our manuscript as the reviewer’s commented. For more detail, please refer to the responses below.

Comment 1: L17: please, replace “the best” by “the most suitable” and add “for Bandafassi” at the end of the objective. L26: What is CI? The reader don’t know yet its meaning yet.

Answer 1: As recommended by the reviewer, we have changed the word in the sentence from ‘‘best’’ by ‘‘most suitable’’ (L14) and we have added “for Bandafassi” (L16) at the end of the objective. 

Comment 2: L26: What is CI? The reader don’t know yet its meaning yet. 

Answer 2: Thank you for pointing this out. The CI means Confidence Interval. We have added it in the abstract section of update manuscript (L25).

Comment 3: L31: What is “with 3-day”? Do you mean with 3-day duration?

Answer 3: Yes, we did mean with 3-day duration. We have now added the word ‘‘duration’’(L30).

Comment 4: L37: remove “current”

Answer 4: We have removed “current” as suggested.

Comment 5: L41:43: please rewrite starting from “three heat waves …. Study periods”. The sentence is misleading. Also in line 43 and 44. I think it should be “A recent study…. summaries that the excess ….. during the three heat …… 2017) showed a 11% ….. increase in risk.”

Answer 5: We accepted the reviewer’s suggestion. We re-wrote the sentence so it starting from “three heat waves …. Study periods” (L43-45). The sentence “A recent study…. increase in risk” has been re-written in the update manuscript (L46-49).

Comment 6: L60: start time and end time.

Answer 6: It is corrected in the revised manuscript (L65).

Comment 7: L119: What do you refer to in this sentence?

Answer 7: We thank the reviewer for raising the issue, we have been expanded the description of the study area also including more information on its climate condition (for example also on the basis of the Köppen climate classification (Cornforth et al., 2019 [45])).

Comment 8: L135:136: move to statistical analysis

Answer 8: We agree with the reviewer’s point of view. We have moved this sentence at the end of statistical analysis subsection (L208-209).

Comment 9: L202: coma is missing “from figure 2, we “

Answer 9: This is corrected (L222).

Comment 10: Figure 2: It is better to re arrange the x-axis to incremental values and add these specific days as vertical lines (using abline in R).

Answer 10: We appreciate this suggestion as it may help to better understand. However, we believe that for methodological consistence it is better to use the dates which major peaks were observed.

Comment 11: L215: Bold is not needed.

Answer 11: As recommended by reviewer, this has been changed in the manuscript (L235).

Comment 12: Page 12: Please, divide into several paragraphs

Answer 12: We have divided the discussion section into several paragraphs in the update manuscript.

Comment 13: L323: Is it the only reason? Is it or due to heat built-up in body?

Answer 13: We thank the reviewer for this observation. It is possible to have others reasons. It may be certainly true. As far as we know, the reason is that, possibly due to their greater physiological adaptation to high temperatures as showed in the manuscript.

Comment 14: L344: correct to “3 days is the most suitable (OR most representative) definition”

Answer 14: As recommended by reviewer, we have changed the word “best” with “most suitable” in the conclusions section (L391).

Reponses to Reviewer #2 comments

We thank the reviewer for careful and thorough reading of this manuscript. The reviewer’s comment help to improve our paper. Please find below point-by-point a detailed reponse to comments as followed.

Comment 1: The heat wave have highest morality rate in cities while this paper only explores the rural heat wave morality. The reason is that heat wave is amplified a lot when considering the cities. I suggest the results from the cities should also be considered.

Answer 1: We appreciate this helpful comment. We agree with this point as showed in the literature review (e.g. Shafiei Shiva et al., 2018 [51]; Basara et al., 2010 [4]; Khan et al., 2019 [33]). The study location was limited to a single rural areas (Bandafassi). We have expanded it in the Data collection subsection. Therefore we will consider this idea of the reviewer as another future possibility. 

Comment 2: What was the base year the heat wave was defined?

Answer 2: We thank the reviewer for raising the issue, the period 1973-2012 was the base year the heat wave.

Comment 3: The heat wave is indeed defined by a temperature, but it does not mean that the threshold temperature should be percentage based. The question is why the percentage-based threshold was used, why not a specific temperature?

Answer 3: We complety agree with the reviewer’s point of view that ‘‘the heat wave is indeed defined by a temperature’’. We found that the both daily minimum and maximum temperature is the most representative variable of daily mortality based the Akaike’s Information Criterion (AIC) (results not show in the manuscript). As highlighted by Chen et al. (2015) [53] heat waves are defined, in general, by temperature indicator, temperature threshold and heat wave duration. In our work, heat waves are defined by (1) both daily minimum and maximum as temperature indicator, (2) relative threshold (87th, 90th, 92th, 95th, 97th percentiles of temperature) as temperature threshold (3) and duration (≥2, ≥3 and ≥ 4) as heat wave duration.

Comment 4: Why only morality data was used why also use the morbidity data also?

Answer 4: We are grateful for this comment. We don’t study the morbidity because our database do not contain the morbidity data. Only daily mortality data are used in this study; see Materials and Methods section ‘‘Daily mortality count data were obtained from the Bandafassi Health and Demographic Surveillance System (HDSS)’’.

Comment 5: What is the correlation coefficient in figure 2? I think it will give some idea between the temperature and death perhaps?

Answer 5: We thank the reviewer for these questions. The relationship between the temperature and the number death were closely correlated because the value of correlation coefficient (0.80) is closely to 1. 

Comment 6: What is the novelty of the study?

Answer 6: We thank the reviewer for raising the issue. As far as we know, it is the first study in Senegal. Poisson generalized additive model (GAM) is used to investigate the effect of heat wave on mortality and distributed lag non-linear model (DLNM) to evaluate the nonlinear association of heat wave definition at different lag days. Heat wave definition using 3 or more consecutive days with both daily minimum and maximum temperature greater than the 90th percentile shows the best model fit. More precisely, our original findings, crucial for climate services, researchers, clinicians, end-users and decision-makers are: we found that females and people aged ≥ 55 years old were at higher risks than males and other different age groups to heat wave related mortality for definition based on 90th temperature percentile with 3-day duration. These results are expected to be useful for decision makers who plan public health measures in Senegal and elsewhere. Climate parameters including temperatures and humidity could be used to forecast heat wave risks over our area study for early warning.

Comment 7: The literature is not complete. Please include the papers on heat wave definition, indices and what heat waves will be in the coming decades:

-Trends in heat wave related indices in Pakistan

-Prediction of heat waves in Pakistan using quantile regression forests

-Spatial distribution of unidirectional trends in temperature and temperature extremes in Pakistan

-Selection of GCMs for the projection of spatial distribution of heat waves in Pakistan

Answer 7: We appreciate the opportunity to include additional references. As suggested by the reviewer, we have included these relevant references in the revised manuscript. 

Paper#1:

-Trends in heat wave related indices in Pakistan; we have added it to in introduction section (Page 3, Line 94-95) and References.

[37] Khan N, Shahid S, Ismail T, Ahmed K, Nawaz N, (2018b) Trends in heat wave related indices in Pakistan. Stoch. Env. Res. Risk A. doi.org/10.1007/s00477-018-1605-2.

Paper#2:

-Prediction of heat waves in Pakistan using quantile regression forests; we have added it to in introduction section (Page 2, Line 81-82; Page 3, Line 89 and Line 94-95) and References.

[28] Khan N, Shahid S, Juneng L, Ahmed K, Ismail T et al. (2019) Prediction of heat waves in Pakistan using quantile regression forests. Atmos. Res. 221, 1–11. doi.org/10.1016/j.atmosres.2019.01.024.

Paper#3:

-Spatial distribution of unidirectional trends in temperature and temperature extremes in Pakistan; we have added it to in introduction section (Page 2, Line 45-46), to in Heat wave definition subsection (Page 5, Line 160) and References.

[8] Khan N, Shahid S, bin Ismail T, Wang, X.-J, (2018a). Spatial distribution of unidirectional trends in temperature and temperature extremes in Pakistan. Theor. Appl. Climatol. 1–15. doi.org/10.1007/s00704-018-2520-7.

Paper#4:

-Selection of GCMs for the projection of spatial distribution of heat waves in Pakistan ; we have added it to in introduction section (Page 3, Line 90 and Line 94-95) and References.

[33] Khan N, Shahid S, Ahmad K, Wang X-J, (2019) Selection of GCMs for the projection of spatial distribution of heat waves in Pakistan. Atmospheric Research 233 (2020) 104688. doi.org/10.1016/j.atmosres.2019.104688.

---

## [Decision Letter · Decision Letter 1]

25 Jan 2021

PONE-D-20-27586R1

Impact of Different Heat Wave Definition on Daily Mortality in Bandafassi, Senegal

PLOS ONE

Dear Dr. Mbaye,

Thank you for submitting your manuscript to PLOS ONE. After careful consideration, we feel that it has merit but does not fully meet PLOS ONE’s publication criteria as it currently stands. Therefore, we invite you to submit a revised version of the manuscript that addresses the points raised during the review process.

We look forward to receiving your revised manuscript.

Kind regards,

Shamsuddin Shahid

Academic Editor

PLOS ONE

Reviewers' comments:

Reviewer's Responses to Questions

**Comments to the Author**

1. If the authors have adequately addressed your comments raised in a previous round of review and you feel that this manuscript is now acceptable for publication, you may indicate that here to bypass the “Comments to the Author” section, enter your conflict of interest statement in the “Confidential to Editor” section, and submit your "Accept" recommendation.

Reviewer #1: All comments have been addressed

Reviewer #2: All comments have been addressed

2. Is the manuscript technically sound, and do the data support the conclusions?

Reviewer #1: Yes

Reviewer #2: Yes

3. Has the statistical analysis been performed appropriately and rigorously? 

Reviewer #1: Yes

Reviewer #2: Yes

4. Have the authors made all data underlying the findings in their manuscript fully available?

Reviewer #1: No

Reviewer #2: No

5. Is the manuscript presented in an intelligible fashion and written in standard English?

Reviewer #1: No

Reviewer #2: Yes

6. Review Comments to the Author

Reviewer #1: I would like to thank the authors for their work in updating the manuscript "Impact of Different Heat Wave Definition on Daily Mortality in Bandafassi, Senegal" based on the 1st round of revision. I recommend minor revision for the 2nd round of the review.

1) L124-127: "The full name of the institutional ... this study.". I believe this is not a correct place for this statement. It should be moved to Acknowledgments.

2) Figure 2 doesn't look professional. I believe the author should correct the x-axis and make an incremental scale and then highlight the peak days by vertical lines. Also the x-axis label should be moved down and not overlap any other text.

3) I believe the author should consider English proofreading service for this manuscript to improve it.

Thank you

Reviewer #2: The authors have addressed all the comments that I made on their manuscript sufficiently. Therefore, I recommend that the manuscript maybe sonsidered.

7. PLOS authors have the option to publish the peer review history of their article (what does this mean?). If published, this will include your full peer review and any attached files.

Reviewer #1: **Yes: **Mohamed Salem Nashwan

Reviewer #2: No

---

## [Author Response · Author response to Decision Letter 1]

25 Feb 2021

February, 2021

Re: Resubmission of “ Impact of Different Heat Wave Definitions on Daily Mortality in Bandafassi, Senegal”, manuscript id: PONE-D-20-27586R1 

Dear Editor :

Thank you for the second round of reviewer’s comments and opportunity to revise our manuscript titled “ Impact of Different Heat Wave Definition on Daily Mortality in Bandafassi, Senegal”. We appreciate the careful reviews and constructive suggestions by the reviewers. The manuscript has substantially improved after making the suggested amendments.

In the following section, find a detailed point-by-point response in red to the reviewers. Changes made in the manuscript are marked using track changes. The revision has been developed in consultation with all co-authors, and each author has given approval to the final version of the manuscript.

Sincerely, 

Mbaye Faye

Saint-Louis, Senegal

Reponses to Reviewer #1 comments 

I would like to thank the authors for their work in updating the manuscript "Impact of Different Heat Wave Definition on Daily Mortality in Bandafassi, Senegal" based on the 1st round of revision. I recommend minor revision for the 2nd round of the review.

We appreciate the reviewer’s comments for the second round of the review. These comments will help us to improve the quality of the manuscript. For more detail, please refer to the responses below.

Comment 1: L124-127: "The full name of the institutional ... this study.". I believe this is not a correct place for this statement. It should be moved to Acknowledgments.

Answer 1: Thank you for noting this. However, during the 1st review process, the editor (Orsolya Voros) recommend that us to insert it into the beginning of the Methods section: 

‘‘Please insert your ethics statement:

‘‘The full name of the institutional review board that approved our specific study is Bandafassi Health and Demographic Surveillance System (HDSS). We confirm that the Bandafassi Health and Demographic Surveillance System (HDSS) approved this study.’’

into the beginning of the Methods section of your manuscript file.’’

However, we have reworded this text in the revised version of manuscript:

‘‘The full name of the institutional review board that reviewed our specific study approval is the Bandafassi Health and Demographic Surveillance System (HDSS). We confirm that this was approved.’’

Comment 2: Figure 2 doesn't look professional. I believe the author should correct the x-axis and make an incremental scale and then highlight the peak days by vertical lines. Also the x-axis label should be moved down and not overlap any other text.

Answer 2: We thank the reviewer for raising the issue. As you suggested, we have now corrected the x-axis of Figure 2 in the revised manuscript.

Comment 3: I believe the author should consider English proofreading service for this manuscript to improve it.

Answer 3: We thank the reviewer for this suggestion. An English proofreading service has been made for the revised manuscript as suggested.

Reponses to Reviewer #2 comments

The authors have addressed all the comments that I made on their manuscript sufficiently. Therefore, I recommend that the manuscript maybe sonsidered.

Thank you for the positive feedback. We appreciate this suggestion which help to improve the readability of our manuscript.

---

## [Editor Report · Decision Letter 2]

15 Mar 2021

Impact of Different Heat Wave Definition on Daily Mortality in Bandafassi, Senegal

PONE-D-20-27586R2

Dear Dr. Mbaye,

We’re pleased to inform you that your manuscript has been judged scientifically suitable for publication and will be formally accepted for publication once it meets all outstanding technical requirements.

Kind regards,

Shamsuddin Shahid

Academic Editor

PLOS ONE

---

## [Editor Report · Acceptance letter]

22 Mar 2021

PONE-D-20-27586R2 

Impact of Different Heat Wave Definitions on Daily Mortality in Bandafassi, Senegal 

Dear Dr. Faye:

I'm pleased to inform you that your manuscript has been deemed suitable for publication in PLOS ONE. Congratulations! Your manuscript is now with our production department. 

Kind regards, 

on behalf of

Dr. Shamsuddin Shahid 

Academic Editor

PLOS ONE